# Independent Predictors of Ovarian Torsion Laterality: Nulliparity (Virgo) and Cyst Presence

**DOI:** 10.3390/life15121819

**Published:** 2025-11-27

**Authors:** Omer Tammo, Aybüke Arıcan, Eda Nur Çetin, Esra Türk Keklik

**Affiliations:** 1Department of Gynecology and Obstetrics, Harran University, Şanlıurfa 63700, Turkey; 2Department of Obstetrics and Gynecology, Harran University, Şanlıurfa 63700, Turkey; 3Şanlıurfa Training and Research Hospital, Şanlıurfa 63700, Turkey

**Keywords:** nulliparity, cyst, ovarian torsion, laterality, hormonal profile, logistic regression, diagnostic predictors

## Abstract

**Objective**: This retrospective cohort study aimed to identify independent clinical predictors specifically associated with the laterality of ovarian torsion, rather than examining potential associations with patient age or serum hormonal profiles. The analysis sought to improve diagnostic accuracy and preoperative risk stratification in ovarian torsion, a gynecologic emergency where fertility preservation is a priority. **Materials and Methods**: Data from 64 patients with surgically confirmed ovarian torsion between January 2018 and June 2025 were retrospectively reviewed at Harran University Faculty of Medicine. Demographic, clinical, ultrasonographic, and hormonal data were collected. Hormonal assays were performed using electrochemiluminescence immunoassays (Roche Diagnostics). Univariate and multivariate logistic regression analyses were employed to identify factors independently associated with torsion laterality. **Results**: The median age of the 64 patients included in the study was 22 years (IQR: 20–33). Right ovarian torsion was detected in 65.6% of the cases. In the final multivariate logistic regression model, nulliparity (VIRGO) (OR: 0.22; 95% CI: 0.07–0.75; *p* = 0.015) and the presence of a cyst (OR: 0.10; 95% CI: 0.02–0.53; *p* = 0.007) were found to be independent and significant predictors of torsion laterality. Both variables demonstrated an effect that reduced the probability of left ovarian torsion. No significant association was found between patient age, parity, other clinical features, and hormonal profiles (progesterone, E2, FSH, LH). The main reason for the lack of backing for the hormonal hypothesis was that hormone measurements in 95.3% of the patients were conducted during the follicular phase, a time at which progesterone and estrogen levels are typically low compared to the luteal phase. **Conclusions**: This study demonstrates that, in addition to the anatomical predisposition for right ovarian torsion, nulliparity (VIRGO) and cyst presence are independent clinical indicators of ovarian torsion laterality. The absence of these features (VIRGO and cyst presence) increases the risk of right-sided torsion. The findings offer valuable information to enhance the index of clinical suspicion in the differential diagnosis of patients presenting with ovarian torsion symptoms. The importance of prompt detorsion and a conservative surgical approach to preserve ovarian viability and secure long-term reproductive health is emphasized.

## 1. Introduction

The twisting of the ovary around its vascular pedicle is known as ovarian torsion (OT), a serious gynecologic emergency that can cause ovarian ischemia and necrosis if it is not quickly diagnosed and treated [1]. Its rarity in emergency departments often contributes to diagnostic delays, frequently resulting in irreversible ovarian damage and the need for oophorectomy [2]. The critical importance of timely intervention underlines the need for a comprehensive understanding of the factors influencing its clinical presentation and progression of the disease [3]. This surgical emergency requires rapid diagnosis and intervention to preserve ovarian function and prevent significant morbidity, particularly loss of fertility [4]. The clinical course and laterality of ovarian torsion can differ depending on demographic characteristics, which are supported by references [5,6].

Research also indicates that right ovarian torsion occurs more frequently. The difference in laterality is due to anatomical factors: the left ovary is more securely fixed by the sigmoid colon, whereas the right ovary’s longer meso-ovarium allows greater mobility, increasing its susceptibility to torsion [7].

Recognizing how hormonal changes over time influence the laterality of ovarian torsion can lead to the creation of more accurate diagnostic methods [8]. This, in turn, can lead to more targeted interventions aimed at preserving ovarian viability, especially in cases where fertility preservation is a priority. Recent research indicates that ovarian dimensions, form, and microscopic characteristics are linked to hormonal variations, and these hormonal fluctuations may impact the likelihood of torsion [9].

The objective of this investigation is to examine the potential connections between the side of ovarian torsion and patient age, as well as hormonal profiles (progesterone, estrogen, FSH, and LH levels). This detailed analysis could enhance diagnostic accuracy and improve preoperative risk stratification. This comprehensive understanding will provide critical insights into the complex interplay of endocrine and anatomical factors, ultimately refining diagnostic paradigms and enabling more proactive intervention strategies.

## 2. Materials and Methods

### 2.1. Study Design and Setting

This retrospective cohort study was designed to investigate the relationship between the laterality of ovarian torsion (OT) and patient age and hormonal profiles. The study was conducted at the Department of Obstetrics and Gynecology at Harran University Faculty of Medicine and included all patients diagnosed with ovarian torsion between January 2018 and June 2025.

### 2.2. Study Population and Data Collection

Data were retrospectively collected from the hospital’s information management system. The medical records of patients meeting the inclusion criteria were reviewed.

### 2.3. Inclusion Criteria

Patients with a diagnosis of ovarian torsion confirmed surgically (via laparoscopy or laparotomy).

Patients whose medical records contained complete age information.

Patients who had serum levels of progesterone, estradiol (E2), follicle-stimulating hormone (FSH), and luteinizing hormone (LH) measured at the time of torsion or within the preceding three months.

### 2.4. Exclusion Criteria

Patients with a prior history of ovarian surgery or pelvic adhesions.

Patients with a history of advanced-stage endometriosis or pelvic inflammatory disease (PID) are known to be at risk of ovarian torsion.

Cases of ovarian torsion associated with pregnancy.

Cases of ovarian torsion developing due to Ovarian Hyperstimulation Syndrome (OHSS).

Patients diagnosed with an ovarian pathology other than ovarian torsion (e.g., malignant ovarian tumor).

Patients with inadequate or incomplete medical records (especially regarding the laterality of ovarian torsion and hormonal data).

### 2.5. Data Variables

The following data were collected and analyzed:

Demographic and Clinical Information: Age (as a continuous variable), parity, presenting complaints (e.g., abdominal pain), time to diagnosis, and prior history of ovarian torsion.

Imaging and Surgical Findings: Ovarian size, presence of a cyst, and Doppler findings reported on ultrasonography. Additionally, the laterality of ovarian torsion (right/left) determined during surgery, the degree of torsion (number of turns), and the appearance of the ovary (signs of ischemia/necrosis) were recorded.

Hormonal Parameters: Serum levels of progesterone, estradiol (E2), FSH, and LH measured at the time of torsion or within the previous three months. Where possible, information on the phase of the menstrual cycle at which the hormones were measured was also recorded.

### 2.6. Hormonal Assay and Doppler Ultrasonography Methodology

Serum levels of progesterone, estradiol (E_2_), FSH, and LH were measured using electrochemiluminescence immunoassay (ECLIA) on a Cobas e 601 analyzer (Roche Diagnostics, Mannheim, Germany). The analytical sensitivities (lower limits of detection) for the assays were as follows: progesterone: 0.03 ng/mL; estradiol: 5.0 pg/mL; FSH: 0.10 mIU/mL; and LH: 0.10 mIU/mL. The intra-assay and inter-assay coefficients of variation (CV) for all hormones were below 5% and below 10%, respectively, in line with the manufacturer’s guidelines.

Doppler ultrasonography was conducted by experienced gynecologists using a Voluson E8 or E10 system (GE Healthcare, Chicago, IL, USA) with a transabdominal (3.5–5 MHz) or transvaginal (5–9 MHz) probe, selected according to the patient’s age and sexual activity. The presence or absence of ovarian blood flow was assessed qualitatively and recorded as ‘Normal’ or ‘Abnormal’ (absent or significantly reduced).

### 2.7. Ethical Approval

Our study was designed in accordance with the Declaration of Helsinki and received ethical approval from the Harran University Ethics Committee (Study ethics committee no. HRÜ/25.14.08; session date: 1.09.2025). The patients included in the study were informed, and their written consent was obtained.

### 2.8. Statistical Analysis

Statistical analysis was performed using SPSS 15.0 for Windows. Descriptive statistics were presented as numbers and percentages for categorical variables, and as median and interquartile range (IQR) for numerical variables. Proportions between groups were compared using the chi-square test. Comparisons of numerical variables (including age) between two independent groups were performed using the Mann–Whitney U test, as normal distribution was not achieved.

Determinant factors were investigated using logistic regression analysis. To identify factors independently associated with torsion laterality, a univariate logistic regression analysis was first performed. Factors with a significance level of *p* < 0.250 in the univariate analysis (age, VIRGO score, parity, history of previous torsion, degree of torsion, presence of a cyst, and estrogen level) were included in the initial multivariate logistic regression model. A stepwise elimination method was applied to this initial model to arrive at the final model. This method ensured the development of the most parsimonious and robust model by sequentially eliminating the least significant variables, retaining only the independent and statistically significant predictors (VIRGO and presence of cyst). The alpha significance level was accepted as *p* < 0.05.

## 3. Results

The median age of the 64 patients included in the study was 22 years (IQR: 20–33). The median age was 21 years (IQR: 19–29.5) for right ovarian torsion cases and 24 years (IQR: 22–35) for left ovarian torsion cases; however, the difference in age between the two groups was not statistically significant (*p* = 0.081). VIRGO positivity was found in 51.6% of patients; this rate was observed as 59.5% in the right torsion group and 36.4% in the left torsion group (*p* = 0.078). No statistically significant difference was found between the right and left groups with respect to other demographic and clinical characteristics, such as parity, presenting complaints, time to diagnosis, previous history of ovarian torsion, ovarian size, Doppler findings, and the degree of torsion, since the *p*-value was greater than 0.05. The presence of a cyst was detected in 71.9% of all cases and was found to be statistically significantly higher in the left torsion group (90.9%) compared to the right group (61.9%) (*p* = 0.014).

### 3.1. Hormonal Findings

When biochemical parameters were examined, progesterone levels were mostly found to be too low to be measured, and no difference was observed between the groups (*p* = 0.596). The primary reason for this critical finding is that 95.3% of the cases studied had hormonal measurements performed during the follicular phase (Table 1). No significant difference was found between the right and left groups for estrogen (E2), FSH, and LH levels either (*p* > 0.05). No statistically significant differences were demonstrated between the right and left torsion groups for any other clinical, ultrasonographic, or hormonal parameters (Table 1).

### 3.2. Univariate and Multivariate Logistic Regression Analyses

Clinical and laboratory variables that might be associated with the laterality of torsion were analyzed using a Univariate Logistic Regression model, with left ovarian torsion as the reference Table 2. The analysis revealed that parity (OR: 1.35; 95% CI: 1.01–1.79; *p* = 0.041), a previous history of ovarian torsion (OR: 0.53; 95% CI: 0.08–0.11; *p* = 0.017), and the presence of a cyst (OR: 0.16; 95% CI: 0.03–0.79; *p* = 0.024) were found to be statistically significantly associated with the laterality of the torsion.

### 3.3. Multivariate Logistic Regression Analysis

To evaluate the independent effect of factors associated with the laterality of torsion, factors identified with a significance level of *p* < 0.250 in the univariate analyses were included in the initial model of the multivariate logistic regression analysis. In the final model derived through the stepwise elimination method, the presence of VIRGO (OR: 0.22; 95% CI: 0.07–0.75; *p* = 0.015) and the presence of a cyst (OR: 0.10; 95% CI: 0.02–0.53; *p* = 0.007) showed an independent and significant association with the laterality of torsion (Table 3).

Both variables demonstrated an effect that reduces the probability of torsion developing on the left side. Variables that showed significance in the univariate analysis, such as parity (*p* = 0.041) and previous torsion history (*p* = 0.017), were not found to be independently significant after controlling for the simultaneous effects of other factors in the multivariate model, and they were subsequently excluded from the final model (all *p* > 0.05).

## 4. Discussion

The etiology of ovarian torsion laterality is shaped by the general consensus that the right ovary is more frequently affected due to anatomical factors [10,11]. However, evidence regarding specific clinical and hormonal predictors that play a role in determining the side of torsion is limited, and strong indicators are required to improve diagnostic accuracy. Recent research, as exemplified by Aiob et al. [12], underscores the need for identifying reliable clinical predictors. In this context, our study’s multivariate analysis results make a significant contribution to the literature by identifying nulliparity (VIRGO) and the presence of a cyst as strong and independent predictors of the laterality of ovarian torsion. Our findings are consistent with existing literature, which reports a higher incidence of right ovarian torsion [10,11]. The common laterality is thought to be caused by anatomical factors, with the left ovary being more firmly anchored by the sigmoid colon, and the right ovary’s longer meso-ovarium allowing it to be more mobile, which can contribute to the condition [13,14].

One of the most notable findings of our study is the emergence of nulliparous status and the presence of a cyst as independent and powerful predictors for the side of ovarian torsion in the multivariate analysis. Retrospective cohort studies have indicated that nulliparous women are more susceptible to ovarian torsion [15,16]. The presence of a cyst is a well-known fact in the literature, significantly increasing the risk of torsion by increasing the weight of the ovary [17].

However, despite the higher cyst prevalence in the raw data for the left ovary, our findings from the multivariate analysis demonstrate that the presence of a cyst reduces the risk of left ovarian torsion. This suggests that the presence of VIRGO (OR: 0.22; 95% CI: 0.07–0.75; *p* = 0.015) and the presence of a cyst (OR: 0.10; 95% CI: 0.02–0.53; *p* = 0.007) retain significant predictive utility for the laterality of torsion, even after controlling for other variables. Specifically, these multivariate regression analysis results indicate that VIRGO status (odds ratio: 0.22) and the presence of a cyst (odds ratio: 0.10) strongly decrease the probability of left ovarian torsion. A lower VIRGO score and the absence of a cyst were, by inference, associated with an increased probability of right-sided ovarian torsion, which is already anatomically predisposed. This suggests that, beyond the anatomical predisposition, the absence of these specific clinical characteristics (VIRGO status and cyst presence) constitutes an additional factor that further elevates the risk of right ovarian torsion. These specific features are considered potentially valuable indicators for the differential diagnosis and management of patients presenting with symptoms of ovarian torsion.

Figure 1 provides a schematic illustration of the proposed mechanism, showing how a cyst and nulliparity might influence the mobility and susceptibility of the ovary to torsion on a specific side.

It is also noteworthy that clinical and demographic variables such as patient age, parity, and ovarian size showed no significant association with laterality, and no significant relationship was found between the hormonal profiles (Progesterone, E2,FSH, LH) and the side of torsion. This finding is consistent with the literature suggesting that hormonal levels may not directly influence the risk or side effects of ovarian torsion but might play an indirect role by leading to cyst formation or ovulation induction [10].

From a clinical perspective, our findings suggest that in a patient presenting with acute pelvic pain, the absence of nulliparity and a detectable cyst should heighten suspicion for right-sided torsion, guiding imaging focus and preoperative planning. Our study found no direct hormonal link, but the strong association with cysts highlights the need to assess and manage benign ovarian conditions, especially in women who have never given birth, as a potential preventative measure against torsion. Future research should explore whether elective cystectomy in select high-risk, nulliparous women could reduce the incidence of ovarian torsion.

Ovarian torsion presents diagnostic challenges, particularly in the pediatric and adolescent populations, where symptoms are often non-specific [18,19]. In this context, conservative management, such as detorsion and cystectomy, is often favored to preserve ovarian function, as ovarian function has frequently been observed to be preserved in post-menarcheal women [20]. Recent findings have cast doubt on traditional views that frequently resulted in oophorectomy based solely on visual appearance, as even ovaries initially appearing dusky or necrotic can recover their full function after detorsion [20,21]. This paradigm shift necessitates a more nuanced surgical approach that prioritizes detorsion and observation over immediate resection [2,22]. Laparoscopic detorsion is often preferred due to its minimally invasive nature and high potential for ovarian salvage [2].

Future research can provide more precise guidelines for clinical practice by examining more closely the mechanisms of independent predictors such as VIRGO status and the presence of a cyst. Furthermore, the development of comprehensive, age-specific diagnostic algorithms that incorporate imaging and clinical predictors could significantly improve early detection.

## 5. Limitations

The retrospective nature and limited sample size of our study restrict the generalizability of the findings. Specifically, the null finding regarding the potential relationship between hormonal profiles and torsion laterality is constrained by scientific limitations. Although hormonal analysis was one of the main objectives of our paper, progesterone levels were mostly found to be too low to be measured in the majority of analyzed patients. Furthermore, 95.3% of the cases had hormonal measurements performed during the follicular phase. This situation severely restricted our ability to evaluate the full potential effects of the hormonal profile, particularly the high levels of progesterone and estrogen anticipated during the luteal phase, on the laterality of ovarian torsion. This constraint constitutes one of the main scientific limitations to the non-support of our original hormonal hypothesis and explains the null finding as a scientific limitation. Future studies will require more comprehensive hormonal analyses based on the phase of the menstrual cycle in which the torsion occurred.

## 6. Conclusions

This study identifies nulliparity (VIRGO status) and the presence of a cyst as independent predictors of ovarian torsion laterality, offering critical clinical indicators in addition to the established anatomical predisposition for right-sided torsion. Presence of VIRGO (OR: 0.22) and a cyst (OR: 0.10) in the multivariate analysis decreased the likelihood of left ovarian torsion, thereby making the absence of these characteristics a factor increasing the risk of right-sided torsion. For clinicians, the absence of nulliparity (VIRGO status) and the absence of a cyst should be regarded as a significant clinical cue that further elevates the risk of right ovarian torsion beyond anatomical inclination. Although our hormonal hypothesis could not be verified due to measurement limitations, the findings provide valuable information to increase the index of clinical suspicion in the differential diagnosis of patients presenting with ovarian torsion symptoms. Regarding management, the importance of rapid detorsion and a conservative surgical approach instead of oophorectomy is once again emphasized to preserve ovarian function and secure long-term reproductive health. Future studies should examine more closely the mechanisms of these clinical predictors and integrate them into diagnostic algorithms.

## Figures and Tables

**Figure 1 life-15-01819-f001:**
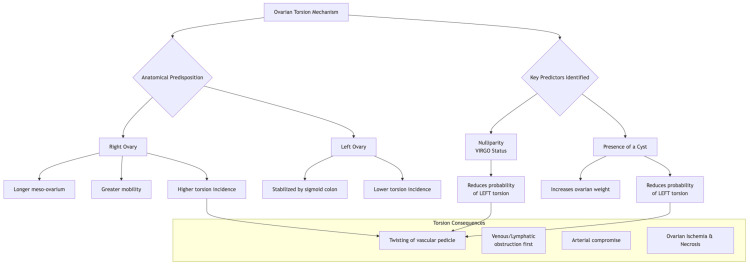
Schematic representation of ovarian torsion pathophysiology and key predictors. The diagram illustrates the anatomical predisposition favoring right-sided torsion due to the longer meso-ovarium and greater mobility of the right ovary, contrasted with the stabilizing effect of the sigmoid colon on the left. It integrates the study’s key findings: Nulliparity (VIRGO status) and Cyst Presence act as independent predictors that significantly reduce the probability of left-sided torsion. The presence of a cyst increases ovarian weight and volume, a key mechanical factor. The final common pathway involves the twisting of the vascular pedicle, leading sequentially to venous and lymphatic obstruction, arterial compromise, and ultimately ovarian ischemia and necrosis if not promptly intervened (Created by the authors for this manuscript).

**Table 1 life-15-01819-t001:** Comparison of demographic, clinical, and laboratory characteristics between right and left ovarian torsion groups.

Characteristic		Total	Right Side *n* = 42 (65.6%)	Left Side *n* = 22 (34.4%)	*p*
**Age Median (IQR)**		22 (20–33)	21 (19–29.5)	24 (22–35)	0.081
**Virgo** ***n*** **(%)**	**No**	31 (48.4)	17 (40.5)	14 (63.6)	0.078
	**Yes**	33 (51.6)	25 (59.5)	8 (36.4)	
**Parity Median (IQR)**		0 (0–2)	0 (0–1)	1 (0–3.3)	0.076
**Presenting Complaint** ***n*** **(%)**	**Lower quadrant pain**	23 (35.9)	15 (35.7)	8 (36.4)	0.959
	**Abdominal pain**	41 (64.1)	27 (64.3)	14 (63.6)	
**Time to Diagnosis (Months)** ***n*** **(%)**	**1 month ago**	16 (25.0)	10 (23.8)	6 (27.3)	0.878
	**2 months ago**	20 (31.2)	14 (33.3)	6 (27.3)	
	**3 months ago**	28 (43.8)	18 (42.9)	10 (45.5)	
**History of Previous Ovarian Torsion** ***n*** **(%)**	**Yes**	3 (4.7)	2 (4.8)	1 (4.5)	0.969
	**No**	61 (95.3)	40 (95.2)	21 (95.5)	
**Ovary Size Max. Diameter (cm) Median (IQR)**		8 (6–10)	8 (6–10.5)	8 (7–10)	0.511
**Cyst Presence** ***n*** **(%)**	**Yes**	46 (71.9)	26 (61.9)	20 (90.9)	**0.014**
	**No**	18 (28.1)	16 (38.1)	2 (9.1)	
**Doppler Findings** ***n*** **(%)**	**Normal**	12 (18.8)	8 (19.0)	4 (18.2)	0.933
	**Abnormal**	52 (81.2)	34 (81.0)	18 (81.8)	
**Degree of Torsion (Number of turns) Median (IQR)**		3 (2–3)	3 (2–3)	2 (2–3)	0.386
**Appearance of Ovary** ***n*** **(%)**	**Necrosis**	7 (10.9)	6 (14.3)	1 (4.5)	0.236
	**Ischemia**	57 (89.1)	36 (85.7)	21 (95.5)	
**Hormone Measurement Time** ***n*** **(%)**	**Follicular phase**	61 (95.3)	40 (95.2)	22 (100)	1.000
	**Luteal phase**	3 (4.7)	2 (4.8)	1 (4.5)	
**Progesterone (ng/mL) Median (IQR)**		0 (0–0)	0 (0–0)	0 (0–0.25)	0.596
**Estradiol (E2) (pg/mL) Median (IQR)**		47 (36–87)	49.5 (37–88.5)	42 (34–91.5)	0.445
**FSH (mIU/mL) Median (IQR)**		4 (4–6)	4 (4–6)	4.5 (3.8–6.3)	0.777
**LH (mIU/mL) Median (IQR)**		9 (8–12)	9 (8–12)	8.5 (7.8–10.3)	0.162

IQR: Interquartile Range, FSH: Follicle-Stimulating Hormone, LH: Luteinizing Hormone, E2: Estradiol.

**Table 2 life-15-01819-t002:** Univariable logistic regression analysis: factors associated with ovarian torsion laterality.

Characteristic	*p*	OR	95% CI Lower-Upper
**Age**	0.136	1.05	0.98–1.13
**Virgo**	0.082	0.39	0.13–1.13
**Parity**	**0.041**	1.35	1.01–1.79
**Presenting complaint (ref: lower quadrant pain) abdominal pain**	0.959	0.97	0.33–2.85
**Time to diagnosis (ref: 1 month ago)**	0.878		
**2 months ago**	0.906	1.08	0.30–3.86
**3 months ago**	0.679	0.77	0.23–2.64
**History of previous torsion**	**0.017**	0.53	0.08–0.11
**Ovary size**	0.857	0.99	0.84–1.15
**Cyst presence**	**0.024**	0.16	0.03–0.79
**Doppler findings (ref: normal) abnormal**	0.933	1.06	0.28–4.00
**Degree of torsion**	0.205	0.74	0.47–1.18
**Appearance of the ovary (ref: ischemia) necrosis**	0.261	0.29	0.03–2.54
**Progesterone**	0.355	1.10	0.90–1.35
**Estradiol (E2)**	0.093	0.56	0.99–1.01
**FSH**	0.384	1.10	0.89–1.36
**LH**	0.514	0.96	0.85–1.09

Ref: reference group; OR: odds ratio; CI: Confidence Interval, FSH: Follicle-Stimulating Hormone, LH: Luteinizing Hormone, E2: Estradiol.

**Table 3 life-15-01819-t003:** Final model of multivariable logistic regression analysis.

Variable	Odds Ratio (OR)	95% Confidence Interval (CI)	*p*
**Virgo presence**	0.22	0.07–0.75	15
**Cyst presence**	0.10	0.02–0.53	7
**Age**	-	-	>0.05
**Parity**	-	-	>0.05
**History of previous torsion**	-	-	>0.05
**Degree of torsion**	-	-	>0.05
**Estradiol (E2) level**	-	-	>0.05

OR: odds ratio; CI: confidence interval.

## Data Availability

The datasets generated and/or analyzed during the current study are available from the corresponding author upon reasonable request.

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
