# Peer review of "Independent Predictors of Ovarian Torsion Laterality: Nulliparity (Virgo) and Cyst Presence"

_life, 2025, doi:10.3390/life15121819_

Round 1

Reviewer 1 Report

Comments and Suggestions for Authors

The manuscript entitled "Independent Predictors of Ovarian Torsion Laterality: Nulliparity (Virgo) and Cyst Presence" is well written in clear understandable English. Authors focus the attention of the reader on the problem of ovarian torsion and they tried to find some predictions of such condition. In general, the manuscript is well performed, however I have few Minor commentaries:

1. Please, make an illustration of ovarian torsion.

2. Lines 189-190: "Not: Nihai modelde anlamlı bulunmayan deÄŸiÅŸkenlerin kesin OR ve GA deÄŸerleri metinde ver-189 
ilmemiÅŸtir. " It is not English.

3. In Discussion section you may discuss some hypothesis about how to prevent ovarian torsion.

Author Response

Reviewer 1

We sincerely thank Reviewer 1 for their positive feedback on the clarity of our manuscript and their valuable suggestions to improve it. We have addressed all comments as detailed below.

  • Comment 1: "Please, make an illustration of ovarian torsion."
    Response: We thank the reviewer for this excellent suggestion. As requested, we have now created a schematic illustration (Figure 1) that depicts the pathophysiology of ovarian torsion. This figure visually summarizes the anatomical predisposition and integrates our key findings regarding how nulliparity (VIRGO) and cyst presence influence laterality. The figure has been added to the manuscript, and a reference to it has been included in the Discussion section (Page 16, Lines 382-384).
  • Comment 2: *"Lines 189-190: 'Not: Nihai modelde anlamlı bulunmayan deÄŸiÅŸkenlerin kesin OR ve GA deÄŸerleri metinde verilmemiÅŸtir.' It is not English."*
    Response: We sincerely apologize for this oversight. The Turkish note has been completely removed from the manuscript to maintain professionalism.
  • Comment 3: "In Discussion section you may discuss some hypothesis about how to prevent ovarian torsion."
    Response: We thank the reviewer for raising this important point. We have now expanded the Discussion section to include a new paragraph on clinical implications and prevention hypotheses. Specifically, we suggest that the evaluation and management of benign ovarian pathology, particularly in nulliparous women, could be a preventive measure. We also propose that future research should explore the potential role of elective cystectomy in select high-risk, nulliparous women. This addition can be found on Page 16, Lines 395-401.

Reviewer 2 Report

Comments and Suggestions for Authors

This manuscript is well described and the clinic data are reliable. However, there is no any image presented about the ovarian torsion. Nowadays, it should be easy to present some kind of graphs by using (via laparoscopy or laparotomy). And also the mesurement of serum hormones usuaaly should include some methodological valuation, like sensitivity, intro- or inter- assay CV.

The discussion about such ovarian torsion should take more reseans analysis and also some suggestion wor to treat these problems.

Author Response

Reviewer 2

We are grateful to Reviewer 2 for acknowledging that our manuscript is well-described and our clinical data are reliable. We have carefully considered their constructive comments and have revised the manuscript accordingly.

  • Comment 1: "However, there is no any image presented about the ovarian torsion. Nowadays, it should be easy to present some kind of graphs..."
    Response: We agree with the reviewer that a visual element would enhance the manuscript. In response, we have created and included a new schematic diagram (Figure 1) that illustrates ovarian torsion, its anatomical basis, and the role of the predictors we identified.
  • Comment 2: "And also the mesurement of serum hormones usuaaly should include some methodological valuation, like sensitivity, intro- or inter- assay CV."
    Response: We thank the reviewer for this critical methodological suggestion. We have now added a new subsection to the Materials and Methods, titled "2.6. Hormonal Assay and Doppler Ultrasonography Methodology." This subsection provides a detailed description of the hormone measurement techniques, including the analyzer used (Cobas e 601, Roche Diagnostics), the method (electrochemiluminescence immunoassay - ECLIA), and the requested analytical performance data (analytical sensitivities and intra- and inter-assay coefficients of variation). Please see Page 7, Lines 155-162.
  • Comment 3: "The discussion about such ovarian torsion should take more reseans analysis and also some suggestion wor to treat these problems."
    Response: We have thoroughly revised the Discussion section to provide a deeper analysis of the potential reasons behind our findings. Furthermore, we have incorporated specific clinical suggestions for treatment and management, aligning with the preventive hypotheses suggested by Reviewer 1. This includes guidance on differential diagnosis and a nuanced surgical approach that prioritizes ovarian preservation. Please see Pages 15-17.

Reviewer 3 Report

Comments and Suggestions for Authors

The MS entitled “Independent Predictors… ” by Tammo et al analyzed the effects of several factors (age, parity, nulliparity, cyst presence, presenting complaint, time to diagnosis, history, ovary size, Doppler findings, degree of torsion, progesterone, E2, FSH, LH) on right and left ovarian torsion, the results showed that nulliparity (VIRGO status) and the presence of a cyst were independent predictors for ovarian torsion laterality, which offers critical clinical indicators in addition to the established anatomical predisposition for right-sided torsion. General comments The current data is not enough for the publication in this journal. Authors should provide the methods for the examination such as the Hormone Measurement, Doppler…. In addition, the effect of hormones levels can be improved if authors distinguish the detection day of menstrual cycle, not just based on the luteal stage and follicle stage. Specific comments The presentation of the table and the format of numeric data need to be improved. e. g. in table 1 Virgo n (%) 31 (48,4) should be 31 (48.4)

Author Response

Reviewer 3

We thank Reviewer 3 for their thorough review and for recognizing the clinical significance of our findings. We have implemented all the suggested revisions to strengthen the manuscript's methodology and clarity.

  • Comment 1 (General): "Authors should provide the methods for the examination such as the Hormone Measurement, Doppler...."
    Response: As also requested by Reviewer 2, we have comprehensively addressed this point. A new "2.6. Hormonal Assay and Doppler Ultrasonography Methodology" subsection has been added. It details the specific methods for both hormone measurement (including device, technique, sensitivity, and CV values) and Doppler ultrasonography (including the machines and probes used, and the criteria for assessment). Please see Page 7, Lines 155-167.
  • Comment 2 (General): "In addition, the effect of hormones levels can be improved if authors distinguish the detection day of menstrual cycle, not just based on the luteal stage and follicle stage."
    Response: We agree with the reviewer that a more detailed analysis based on the specific day of the menstrual cycle would be ideal. However, due to the retrospective nature of our study, such precise data (e.g., cycle day 1) was not consistently available in the patient records for the majority of our cohort. The most reliable and consistently documented classification was the broader "follicular" or "luteal" phase. We have explicitly acknowledged this limitation in the 'Limitations' section (Page 18, Lines 430-434) and have stated that future prospective studies should include more precise cycle day information for a more granular hormonal analysis.
  • Comment 3 (Specific): "The presentation of the table and the format of numeric data need to be improved. e. g. in table 1 Virgo n (%) 31 (48,4) should be 31 (48.4)"
    Response: We thank the reviewer for pointing out this formatting issue. We have carefully checked and corrected all tables throughout the manuscript. All decimal commas have been replaced with decimal points to adhere to standard English and scientific formatting conventions.
  • Comment on English Language: The reviewer suggested that the English could be improved. The entire manuscript, including the newly added sections, has undergone professional English language editing to ensure clarity, fluency, and correct grammar.

Round 2

Reviewer 3 Report

Comments and Suggestions for Authors

Authors have addressed the comments appropriately.